# In Vivo Microelectrode Arrays for Detecting Multi-Region Epileptic Activities in the Hippocampus in the Latent Period of Rat Model of Temporal Lobe Epilepsy

**DOI:** 10.3390/mi12060659

**Published:** 2021-06-03

**Authors:** Yuchuan Dai, Yilin Song, Jingyu Xie, Shengwei Xu, Xinrong Li, Enhui He, Huabing Yin, Xinxia Cai

**Affiliations:** 1State Key Laboratory of Transducer Technology, Aerospace Information Research Institute, Chinese Academy of Sciences, Beijing 100190, China; daiyuchuan18@mails.ucas.edu.cn (Y.D.); ylsong@mail.ie.ac.cn (Y.S.); xiejingyu16@mails.ucas.ac.cn (J.X.); swxu@mail.ie.ac.cn (S.X.); lxr8118@126.com (X.L.); heenhui17@mails.ucas.ac.cn (E.H.); 2School of Electronic, Electrical and Communication Engineering, University of Chinese Academy of Sciences, Beijing 100049, China; 3Division of Biomedical Engineering, University of Glasgow Room 626, Rankine Building, Oakfield Avenue, Glasgow G12 8LT, UK; Huabing.Yin@glasgow.ac.uk

**Keywords:** multi-region detection, implantable probe, temporal lobe epilepsy, rhythmic oscillations, hippocampus

## Abstract

Temporal lobe epilepsy (TLE) is a form of refractory focal epilepsy, which includes a latent period and a chronic period. Microelectrode arrays capable of multi-region detection of neural activities are important for accurately identifying the epileptic focus and pathogenesis mechanism in the latent period of TLE. Here, we fabricated multi-shank MEAs to detect neural activities in the DG, hilus, CA3, and CA1 in the TLE rat model. In the latent period in TLE rats, seizures were induced and changes in neural activities were detected. The results showed that induced seizures spread from the hilus and CA3 to other areas. Furthermore, interneurons in the hilus and CA3 were more excited than principal cells and exhibited rhythmic oscillations at approximately 15 Hz in grand mal seizures. In addition, the power spectral density (PSD) of neural spikes and local field potentials (LFPs) were synchronized in the frequency domain of the alpha band (9–15 Hz) after the induction of seizures. The results suggest that fabricated MEAs have the advantages of simultaneous and precise detection of neural activities in multiple subregions of the hippocampus. Our MEAs promote the study of cellular mechanisms of TLE during the latent period, which provides an important basis for the diagnosis of the lesion focus of TLE.

## 1. Introduction

Implantable microelectrode arrays (MEAs) are capable of detecting neural activity at the single-cell level and are widely applied to the study of the functional circuitry of neuronal networks in health and disease. MEAs with multiple probes allow the simultaneous detection of different areas in the brain. The development of MEAs promoted the study of neural circuit mechanisms of epileptogenesis [1,2] and the novel treatment of intractable epilepsy [3,4]. 

Temporal lobe epilepsy (TLE) is the most common form of focal epilepsy, and it can lead to severe brain disorders, including long-term impaired mood and cognitive function [5,6]. The pathogenesis underlying TLE is complicated, and resistance to drugs results in an increasing number of patients with intractable epilepsy [7]. Therefore, the mechanism of epileptogenesis, especially at the cellular level, needs to be further explored.

Previous evidence revealed that the lesion focus of TLE is mainly concentrated in the hippocampus [8]. The evolution of TLE is characterized by several stages, including the latent period and the chronic period. Patients and animal models with an initial precipitating injury often experienced a seizure-free time interval known as the “latent period” [9]. To explore the epileptic focus and epileptogenesis mechanism in the formation of epilepsy, growing research has focused on the alteration of brain function and the neural circuit in the latent period after brain insults [10,11]. Because the lesion focus and its mechanisms at the cellular level in the latent period of TLE are not well understood, the customized design of an MEA optimized for multi-region in vivo detection is urgently needed to explore the neural dynamics between the hippocampal subregions during this period.

In the present study, we designed a silicon MEA, for which the detecting sites were distributed corresponding to the cell layers of the hippocampus. We applied the MEA to carry out simultaneous detection in multiple subregions of the hippocampus in the latent period of the TLE rat model. Seizures were induced, and the changes in discharge activities of neurons in different subregions were assessed. The neural signals in normal rats before epileptic modeling were also recorded as controls. We aimed to design novel MEAs to explore the changes in electrophysiological function at the cellular level and determine the accurate location of the lesion in the latent period of epilepsy.

## 2. Materials and Methods

### 2.1. Regent and Apparatus

Lithium chloride anhydrous (LiCl), pilocarpine hydrochloride, and scopolamine methyl bromide were purchased from Macklin Biochemical Corporation (Shanghai, China). Saline (0.9% NaCl) was obtained from Shuanghe Corporation (Beijing, China). Phosphate buffered saline (PBS, 0.1 M, Ph 7.4) was from Sigma (Shanghai, China).

The interface modification of the microelectrode array was carried out on an electrochemical workstation (Gamry Reference 600, Gamry Instruments, Warminster, PA, USA). Animals in the present study were anesthetized by an isoflurane anesthesia machine (RWD520, RWD life science, Shenzhen, China).

### 2.2. Design of the Microelectrode Array for Multi-Region Detection

In the present study, a silicon, four-shank (32-channel) microelectrode array (MEA) was designed and fabricated. To simultaneously detect the neural signals in different cell layers of the hippocampus, the shape of the MEA was designed in correspondence to the anatomical structure of the hippocampus (Bregma –3.6 mm, Figure 1a). Each shank had 8 detecting electrodes capable of detecting neural signals in two layers (Figure 1b). The site diameter was 15 μm, and the interval between sites was 30 μm; this design matched the size and distribution density of the hippocampal neurons. To confirm whether MEAs were capable of multi-region detection, the red fluorescent dye DiI was smeared on the back of the MEAs, and the implantable position of the MEAs in the hippocampus was confirmed by postmortem histochemistry after in vivo detection of neural activities of epileptic rats (Figure 1c).

### 2.3. Fabrication of the MEA

Silicon is characterized by low cytotoxicity and excellent biocompatibility. Micro-scale silicon demonstrated good toughness and could withstand the stress of brain tissue during the in vivo detection [12]. In the present study, silicon was chosen as the electrode substrate, and MEAs were fabricated on a silicon-on-insulator (SOI) wafer (25 μm Si; 1 μm SiO_2_; 550 μm Si) using micro-electro-mechanical system (MEMS) technology [13]. Initially, the surface of the SOI wafer was thermally oxidized to produce a thin SiO_2_ film (Figure 2a) to insulate the surface of the SOI wafer. The wafer was then spin-coated with photoresist for photoetching the conducting areas (Figure 2b). The conducting layer was composed of titanium/platinum (Ti/Pt, 30 nm/250 nm) and was deposited on the SiO_2_, followed by lift-off processing to form detection sites, bonding pads, and microwires (Figure 2c). In the present study, platinum was used for the conducting interface of the MEAs due to its chemical stability, good biocompatibility, and high conductivity [14,15]. The titanium layer was deposited to enhance the adhesion between the Pt layer and the SiO_2_ layer [16]. Thereafter, an insulating layer of 500 nm Si_3_N_4_/300 nm SiO_2_ was deposited (Figure 2d) and was then selectively etched to expose detecting electrodes and bonding pads (Figure 2e). The strategy of co-deposition of Si_3_N_4_ and SiO_2_ was used to prevent the bending of MEAs due to mechanical stress [17]. The shape of the MEA was formed by deep etching of the top insulating layer and silicon layer (Figure 2f). Before the back wet etching of the SOI wafer, the top surface was spin-coated with the negative photoresist and black adhesive (Figure 2g). The SOI wafer was then boiled in KOH (50%, 80 °C) and thoroughly cleaned to release the individual MEA (Figure 2h). Finally, the MEA was welded on the printed circuit board (PCB) and insulated with silicone rubber (Figure 2i). Ultrasound in the acetone was performed to ensure that the sites of the MEA were clean (Figure 2j). To decrease the electrode impedance and enhance the signal/noise ratio for recording, the detecting sites were electroplated with platinum nanoparticles (PtNPs) via chronoamperometry. PtNPs can enlarge the specific surface area and enhance the electron transmission capabilities of detecting sites. In addition, electroplated PtNPs are characterized by chemical stability and strong mechanics, which is conducive to stable and durable electrophysiological detection [13]. The electroplating solution was made by mixing 48 mM H_2_PtCl_6_ and 4.2 mM Pb(CH_3_COO)_2_ in a volume ratio of 1:1. Electroplating was carried out at −1.2 V for 50 s using the three-electrode system with a Ag/AgCl reference electrode and a Pt counter electrode. The electrode impedance was assessed by electrochemical impedance spectroscopy (EIS) in PBS.

### 2.4. Lithium-Pilocarpine Induced Epileptic Rat Model

Male Sprague-Dawley rats (250–300 g) were obtained from Vital River Laboratory Animal Technology Co., Ltd (Beijing, China) and used to establish TLE models in the present study. All rats were housed individually under a controlled environment with a 12-h light/dark cycle. Temperature and humidity were maintained at 22 ± 2 °C and 40 ± 5%, respectively. All experiments were performed with the permission of the Beijing Association on Laboratory Animal Care, license number SYXK(JING)2020-0045.

Initially, LiCl (127 mg/kg) was intraperitoneally (i.p.) injected into rats. After 18–20 h, scopolamine (1 mg/kg, i.p.) was injected to alleviate cholinergic damage to the peripheral nerve system, followed by the injection of pilocarpine (10 mg/kg, i.p.) 30 min later. Rats that demonstrated fourth or fifth Racine stages within 30 min after pilocarpine injection were considered successful epileptic models. Epileptic rats were then housed to recover for 48 h and were subject to the following experiments when they entered the latent period.

### 2.5. In Vivo Recording of Neural Signals in TLE Rat Models

Rats were fixed on a stereotaxic device and received craniotomies under isoflurane anesthesia throughout the procedure. The concentration of isoflurane was 4−5% for induction and 0.5–2% for maintenance of general anesthesia. At the same time, a 37 °C heating pad was used to maintain the body temperature of the rats. After the craniotomy, the MEA was slowly implanted into the hippocampus (AP −3.6 mm) from the left hemisphere according to the rat brain atlas of Paxinos and Watson. A stainless-steel nail was anchored on the skull above the cerebellum for connection with the ground wire. The red fluorescent dye DiI was smeared on the back of the MEA to confirm the location of the detecting sites after the experiment.

In the present study, hippocampal signals in normal rats (n = 3) without epileptic modeling (normal) were obtained as controls. For rats in the latent period of epilepsy (n = 3), baseline signals (resting state, RS) were recorded for 15 min before the induction of seizures. Thereafter, the pilocarpine (3 mg/kg, i.p.) was injected into rats to induce seizures. The period after induction was divided into three stages: the beginning stage (BS, first 15 min after induction), middle stage (MS, second 15 min after induction), and grand mal stage (GS, 30 min after induction). Changes in neural electrophysiology during these stages were detected and compared. During the experiment, the electrophysiological signals of hippocampal neurons were monitored by the MEA at a sample rate of 20 kHz with Blackrock Microsystems (HTRP-128, Salt Lake City, UT, USA). Before signal recording, the instrument was run for at least 10 min until the background noise was stable. The threshold for neural spike detection was set to 3 times the maximum amplitude of the background noise. Neural spikes were then extracted by high-pass filter (>250 Hz) and local field potentials (LFPs) were obtained by low-pass filter (0–250 Hz). Recorded neural signals were analyzed after in vivo experiments, and rats received transcardial perfusion of paraformaldehyde (PFA, 4%) followed by histochemical analyses of the tissue implanted with the MEA.

### 2.6. Data Analysis and Statistics

Spike sorting, spike firing rate, spike autocorrelogram, and power spectral density (PSD) analyses of the recorded neural spikes and LFPs data were carried out using the Offline Sorter and Neuroexplorer (Plexon Inc, Dallas, TX, USA). For neural spike sorting, the spike features were extracted using principal component analysis (PCA), and the first three principal components were used for separating single-unit spikes using the Valley-Seeking algorithm and T-distribution E-M algorithm. Each single-unit spike was considered as an action potential train from a neuron. The peak–valley duration, symmetry, and mean of the autocorrelograms of the spike waveforms were then calculated for clarification of single-unit spikes by K-means cluster analysis, which determined whether they were discharged by excitatory neurons (principal cells) or inhibitory neurons (interneurons). According to physiological criteria, principal cells are characterized by longer spike duration and less asymmetry compared with interneurons. Moreover, the autocorrelograms of principal cells exhibit a fast decay, whereas those of interneurons usually show a slow decay. 

Statistical analyses and graphs were performed with Prism 8 (GraphPad, San Diego, CA, USA) and Origin 8 (OriginLab, Northampton, MA, USA). Data were calculated as means ± SEM. The mean values were compared by a two-tailed *t*-test for two groups. ANOVA followed by Tukey’s post hoc test or a Kruskal-Wallis test followed by Dunn’s post hoc test was used for multiple groups. A statistical significance of *P* < 0.05 was set for all analyses.

## 3. Results

### 3.1. Morphology and Impedance Test of the MEAs

In the present study, MEAs were surface-modified with platinum nanoparticles (PtNPs) to decrease electrode impedance. Bare detecting sites were smooth and tidy (Figure 3a), and they became dark and raised after modification (Figure 3d). MEAs were observed with scanning electron microscopy (SEM). PtNP detecting sites were rough and distributed with dense particles compared with bare sites (Figure 3b,e). The image at high magnification showed that PtNP detecting sites had complex and porous structures (Figure 3c,f), which could enlarge the specific surface area and attachment ability to brain tissues. The impedance of the detecting sites was assessed via EIS from 100 kHz to 0.2 Hz. The results showed that modification with PtNPs reduced the impedance magnitude (Figure 4a) and improved the phase shift of the detecting sites (Figure 4b). The average impedance magnitude of detecting sites at 1 kHz decreased from 7125.89 ± 900.7 Ω/μm^2^ to 62.30 ± 3.28 Ω/μm^2^ (Figure 4c), and the phase angle decreased from –72.98 ± 1.65° to –13.93 ± 1.36° (Figure 4d).

### 3.2. Spatiotemporal Variations of Neural Spikes in Multiple Hippocampal Subregions

Figure 5a,b show the variations of neural spikes and local field potentials (LFPs) detected by the fabricated MEAs. Compared with the normal stage, the LFPs fluctuated, accompanied by spinous waves in the RS. After seizure induction, the neural spikes changed rapidly, becoming dense and fast. Spinous waves in the LFPs disappeared in the BS, and the amplitudes of LFPs were significantly increased in the MS. The neural spike frequency and LFP fluctuation reached their maximums in the GS, approximately 30 min after induction.

The differences in firing rates of neural spikes were compared between areas. As shown in Figure 6a, no significant differences were observed between areas in the normal stage. In the RS of epileptic rats, neurons in CA3 fired at the highest frequency, whereas neurons in the hilus fired at the lowest frequency, compared with other areas. After the induction of seizure, neurons in all areas were gradually excited along with the evolution of the seizure, and the firing rate of neurons in the hilus area rose the most rapidly. During the BS, the mean firing rate of hilus neurons increased but was still lower than that of CA3 neurons. Subsequently, the neural firing rate in the hilus continued to increase and reached the level of that in the CA3 area. In the MS, the difference between neural firing rates in the hilus and CA3 was not statistically significant. During the GS, the mean firing rate of neurons in hilus became highest compared with neurons in other areas. In addition, the neural firing rates in the CA3 area were always significantly higher than those in the DG area during all resting state and seizure stages. The contours were then fitted to analyze the distribution of spike power. The results showed that there were no significant differences between the spike power in the four subregions in the normal stage (Figure 6b) or the RS (Figure 6c). With the induction of seizure, the spike power in the hilus and CA3 rose significantly (Figure 6d), followed by a rise in spike power in the DG and CA1 (Figure 6e,f). This implies that the seizure activities of neurons spread from the hilus and CA3 to other areas. Thus, it can be inferred that the epileptic focus was located at the hilus and CA3 in the latent period of TLE.

### 3.3. Neural Spikes of Interneurons and Principal Cells in Epileptic Focus

Neurons in the hippocampus mainly consist of principal cells and interneurons. To explore what kind of neurons caused the above changes, the neural spikes from principal cells and interneurons were classified (Appendix A in Appendix A). 

As shown in Figure 7a, the firing rate of principal cells in the four subregions was lower in the RS compared with the normal stage. For interneurons, their activities in the DG and CA1 were decreased, whereas those in the hilus and CA3 remained high, especially in the hilus in the RS. After the seizure induction, the firing rates of principal cells and interneurons were significantly increased in all areas of the hippocampus with the aggravation of seizure. Furthermore, the firing rates were compared between principal cells and interneurons. Firing rates of both types of neurons displayed no differences in the normal stage. However, in the BS and MS, the firing rates of interneurons were significantly higher than those of principal cells in the hilus and CA3 areas. Although the differences between interneurons and principal cells only showed statistical significance in the CA1 area in the MS, the firing rates of interneurons in the hilus and CA3 remained higher than those of principal cells. In the GS, both types of neurons became overexcited. However, the firing rates of interneurons were much higher than those of principal cells in the hilus. It can be inferred that the hyperexcited activities in the hilus and CA3 areas were dominated by interneurons.

### 3.4. Synchronized Neural Activities and Rhythmic Oscillations of Interneurons

The spread of seizures was usually characterized by the synchronized activities of neurons. At the cellular level, an autocorrelogram analysis at 200 ms was carried out for principal cells and interneurons of the hilus and CA3 areas in the GS. The results showed rhythmic oscillations of interneurons in both subregions, whereas principal cells demonstrated few oscillations (Figure 7b). In addition, the oscillation frequency of interneurons was about 15 Hz. Our results suggest that the rhythmic oscillations of interneurons were involved in seizure evolution. In addition, the power spectral densities (PSD) of spikes and LFPs were subsequently jointly analyzed. The results showed that in all hippocampal areas, the power of LFPs was mainly focused on the frequency band of 1–3 Hz in the normal stage and 4–8 Hz in the RS, whereas the PSD curve of spikes showed no obvious peak (Figure 8a,b). During the BS, a peak occurred near the 20 Hz frequency band of the spike PSD curve in the hilus, CA3, and CA1 areas (Figure 8c). In addition, the peaks of the LFP PSD curves in the DG, hilus, and CA3 areas began to shift to the higher frequency band of 10–20 Hz, whereas the curve peak in CA1 was still at a low-frequency band. In addition, the frequencies corresponding to the peaks of spike PSD and LFP PSD in the CA3 were similar. Subsequently, the PSD curve peaks between the spikes and LFPs were similar to each other at the frequency band of 10–20 Hz in the MS (Figure 8d) and finally coincided at the frequency band of 9–15 Hz (Figure 8e), which indicated the synchronous activities of neurons after the seizure induction. Because the synchronized frequency band (9–15 Hz) of the LFPs and spikes was approximately the same as the oscillation frequency (about 15 Hz) of the interneurons, the results imply that the dynamics of the neuron population were mainly dominated by rhythmic oscillations of interneurons in the hilus and CA3 areas, which might cause the formation and spread of seizure activities.

## 4. Discussion

In this study, multi-shank MEAs were designed for multi-region recording of neural activities in the hippocampus in the latent period of the TLE rat model. The results showed that neurons in the hilus and CA3 regions were highly excited during the latent period. Meanwhile, neurons in both regions were more sensitive to induction stimulation and became the most excited compared with neurons in other areas. Spike power contours further indicated that the seizure activities spread from the hilus and CA3 to other areas. These results suggest that the epileptic focus was at the hilus and CA3 in the latent period of TLE. In addition, the characteristics of different types of neurons were further analyzed to explore the cellular mechanisms. Rhythmic oscillations at 15 Hz of interneurons were observed in the hilus and CA3 areas in grand mal seizures. In addition, the power distributions of spikes and LFPs gradually synchronized in the 9–15 Hz frequency domain with the aggravation of a seizure. It can be deduced that the synchronized neural activities during seizures were driven by the oscillations of interneurons. 

In the present study, the fabricated MEAs promoted the precise detection of neural activities in different subregions of TLE during the latent period. The application of multi-shank MEAs contributed to the diagnosis of lesion sites and the exploration of the cellular mechanisms of epilepsy. Previous studies revealed that neuron function in multiple subfields of the hippocampus played a critical role in epileptic pathogenesis [18,19]. To study the roles of these subfields in epileptogenesis mechanisms, it is important to choose an experimental paradigm, whether in vivo or in vitro. In vitro MEAs have been used to explore the mechanisms of ion channels, network connectivity, and information flow during epileptiform activities in acute hippocampal slices and cultured neurons [20,21]. However, brain slices and cultured neurons are insufficient to reflect the intact neural network activities in the brain and cannot be applied for long-term detection. The strategy used in the present study has more advantages for exploring the mechanisms of living neural networks and the real-time physiological status in epilepsy.

The neural activities detected by fabricated MEAs indicated that the neurons in different subfields had distinct susceptibility to risk factors of epilepsy. In a previous study, histochemical analysis revealed that cell damage after epileptic modeling first took place in the DG and hilus areas, followed by CA3 and CA1 [22]. A recent study used tungsten wire electrodes to detect electroencephalogram (EEG) activities in multiple brain regions in the pilocarpine-induced status epilepticus model. The results showed robust EEG activities in the DG and CA3 areas, and relatively moderate responses in the CA1 area during status epilepticus [23]. The present study further discovered that on the cellular level, the neurons in the hilus and CA3 were more vulnerable to epileptic risk factors, and the interneurons in these areas were more injured by epileptic modeling. The results address the advantages of implantable multi-shank MEAs in studying the micro-mechanisms of neural networks between subregions in TLE.

In the present study, the spinous waves were observed in the LFPs during the resting state of the epileptic latent period. In addition, abnormal neural activities in the hilus and CA3 were observed in this period. However, such phenomena did not occur in the normal stage. This suggests that disorders in the electrophysiological function of the hippocampus took place a short time (48 h) after the epileptic modeling, resulting in the abnormal discharges of neural spikes and LFPs even in the absence of a seizure. It also implies that the early neural damage might first cause disorders in cellular function, which can be judged and predicted by electrophysiological activities. In the present study, the dysfunction in the hippocampal subregions was successfully detected at the cellular level, which is conducive to the prediction of epileptic formation and the accurate diagnosis of the seizure focus. Furthermore, studies have been devoted to the targeted and closed-loop regulation of epilepsy in recent years [24,25], and they are highly dependent on accurately detecting the epileptic focus. The fabricated MEAs and novel findings in the present study could provide an important basis for developing targeted therapy for epilepsy in the future.

Another important finding in the present study was that the spikes and LFPs were gradually synchronized in frequency domain after seizure induction. LFPs are derived from the action potentials and membrane potential-driving fluctuations of local neurons. In addition, the power distribution of LFPs is closely related to the composition of active neuron types. A previous study indicated that the shift from a lower frequency to a higher frequency of LFP power was due to the synchronous activities of neurons [26]. In the present study, autocorrelogram analyses demonstrated strong oscillations of interneurons in the hilus and CA3 areas during grand mal seizures. In addition, the synchronized frequency of PSD was consistent with the oscillation frequency. Considering the hyperexcitability of interneurons in the hilus and CA3 during seizures, it can be inferred that the shift in the frequency domain of LFP power was related to the periodic oscillation of interneurons. Therefore, our results indicate that the fluctuations of LFPs followed the changes of neuronal activities, and the peak in frequency depended on the rhythm of neural oscillations in the present study.

The neural network in the hippocampus consists of excitatory principal cells and inhibitory interneurons [27]. Interneurons play a critical role in the modulation of the balance of excitation and inhibition in the hippocampus, and a disturbed balance has been found to underlie epileptogenesis [28]. In addition, previous studies demonstrated that interneurons were involved in the generation of hippocampal theta activities (4–8 Hz) [29,30], which were proven to be important for information processing and spatial cognition [31]. However, in the present study, the interneurons were more excited and sensitive than principal cells, and they rhythmically oscillated at a higher frequency band (9–15 Hz) after seizure induction. It can be deduced that epileptic modeling caused the injury in interneuron function, resulting in their hyperexcitability and disrupting their inherent discharge rhythm. This might explain why the hilus and CA3 were more susceptible to risk factors and why the induced seizure started in these two areas. Moreover, the changes in the oscillation rhythm of interneurons might serve as a potential biomarker for the diagnosis of epilepsy.

## 5. Conclusions

In the present study, multi-shank microelectrode arrays were designed and fabricated to detect the neural activities in multiple hippocampal regions in the latent period of TLE. The results showed that neuronal dysfunction was mainly focused in the hilus and CA3 areas after epileptic modeling, and interneurons in these areas were more impaired than principal cells. Furthermore, rhythmic oscillations of interneurons characterized by higher frequencies (alpha frequency band) were found in the latent period of epilepsy. The application of our MEAs revealed the cellular mechanism in different hippocampal subregions of TLE and provides an important basis for the diagnosis of the lesion focus in the latent period of TLE. Our findings also form the basis for targeted treatment of epilepsy in the future.

## Figures and Tables

**Figure 1 micromachines-12-00659-f001:**
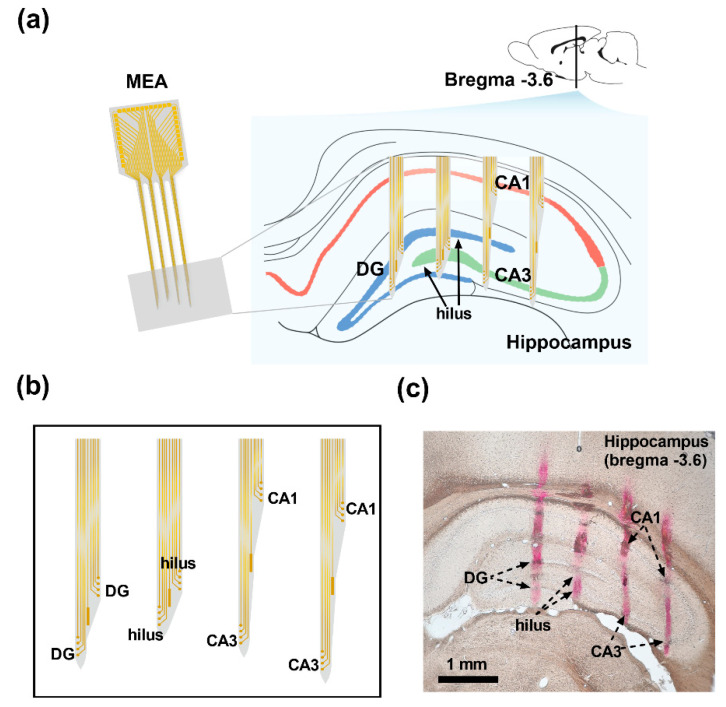
Illustration for the design of the MEA and the implantable position during the in vivo electrophysiological experiment. (**a**) The shape of the MEA. (**b**) Detecting sites were distributed in correspondence to the cell layers of the hippocampus. (**c**) Postmortem histochemistry. The red traces indicate the shanks of the implanted MEA.

**Figure 2 micromachines-12-00659-f002:**
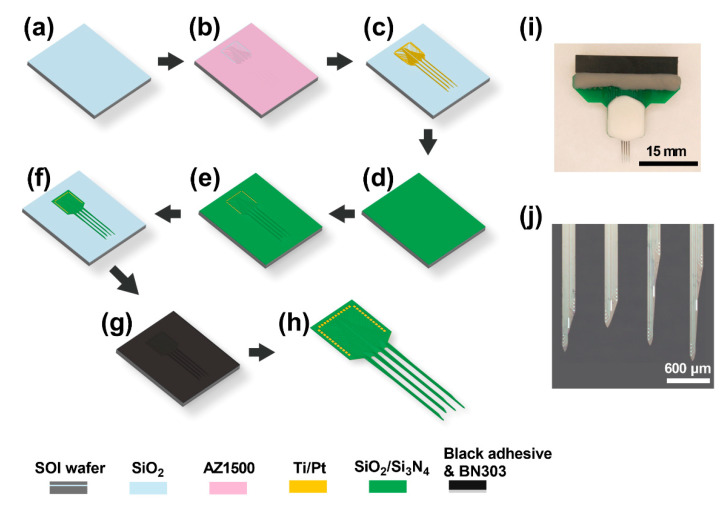
Schematic illustration of the fabrication of the MEA. (**a**) The SiO_2_ film was produced by thermal oxidation. (**b**) Spin-coating of photoresist for photoetching the conducting areas. (**c**) Metallization and lift-off of the Ti/Pt conducting layer. (**d**) Deposition of the insulating layer of Si_3_N_4_/SiO_2_. (**e**) Selectively etching of the insulating layer. (**f**) Plasma deep etching. (**g**) Spin-coating of the negative photoresist and black adhesive. (**h**) Release of MEA by back wet etching. (**i**) Packaged MEA. (**j**) Cleaned MEA after ultrasound in acetone.

**Figure 3 micromachines-12-00659-f003:**
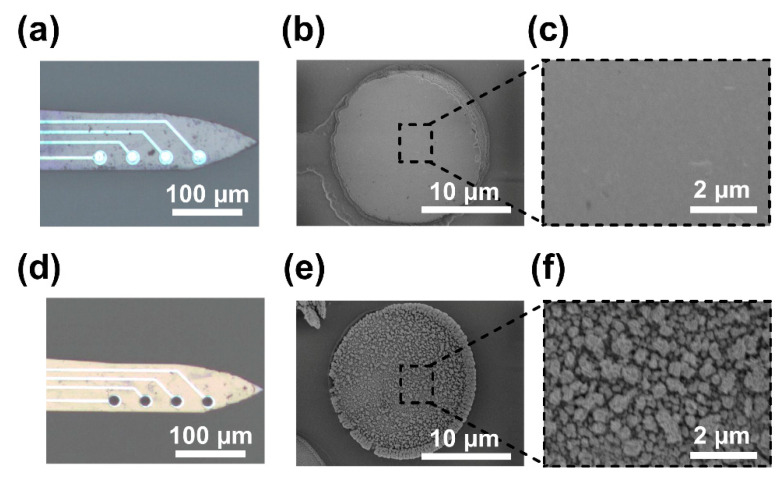
Surface modification of detecting sites. (**a**) Unmodified electrode. (**b**) and (**c**) The bare site observed in scanning electron microscopy (SEM) at 3.5kx and 20kx magnification, respectively. (**d**) The detecting site after modification with PtNPs. (**e**) and (**f**) The PtNP-modified site observed in SEM at 3.5kx and 20kx magnification, respectively.

**Figure 4 micromachines-12-00659-f004:**
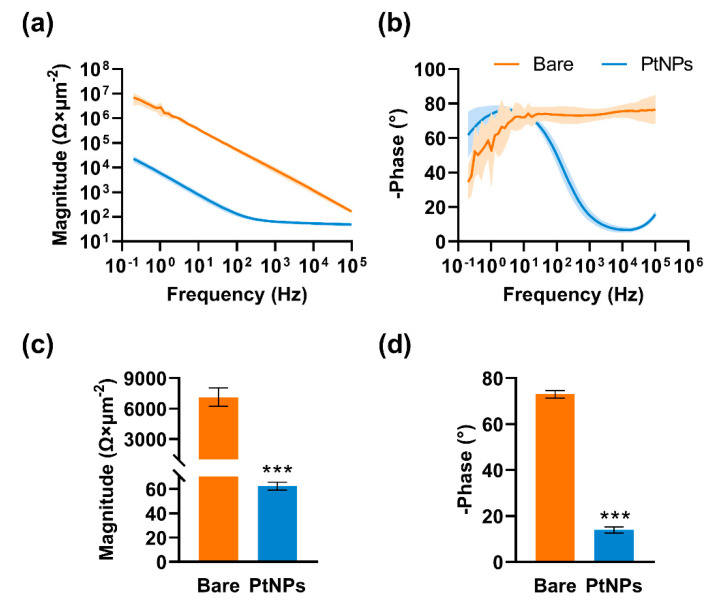
Impedance magnitude and phase assessment. (**a**) and (**b**) Electrode impedance magnitude and phase distribution before and after PtNP modification. (**c**) The average impedance magnitude of electrodes at 1 kHz decreased from 7125.89 ± 900.7 Ω/μm^2^ to 62.30 ± 3.28 Ω/μm^2^. (**d**) The phase changed from –72.98 ± 1.65° to –13.93 ± 1.36°. *** *P* < 0.001, paired *t*-test, n = 10 for each group.

**Figure 5 micromachines-12-00659-f005:**
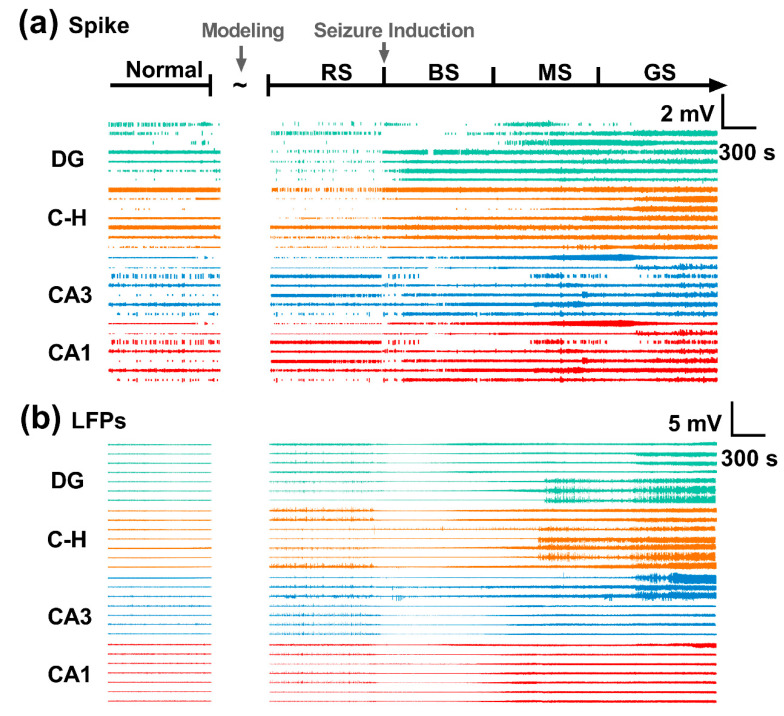
In vivo recording of neural electrophysiology. (**a**) Neural spikes and (**b**) local field potentials (LFPs) during the normal stage and the resting state (RS), beginning stage (BS), middle stage (MS), and grand mal stage (GS) of seizure. C-H: hilus.

**Figure 6 micromachines-12-00659-f006:**
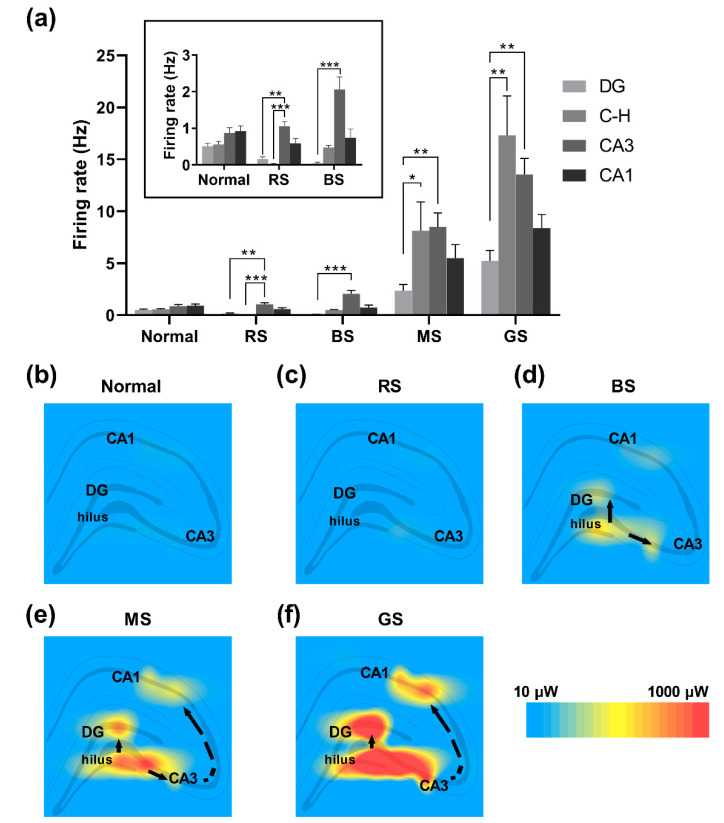
Changes in firing rates of neurons in the hippocampus. (**a**) The comparison of the firing rate of neurons between subfields in the normal stage and the resting state (RS), beginning stage (BS), middle stage (MS), and grand mal stage (GS) of seizure (n = 12 recorded channels for each group) (**b**–**f**) Power contours of neural spikes in each stage. The chart in the box in (**a**) illustrates an enlarged view of firing rates in the normal stage, resting state, and beginning stage of a seizure. * *P* < 0.05, ** *P* < 0.01, and *** *P* < 0.001, Kruskal-Wallis test followed by Dunn’s post hoc test. C-H: hilus.

**Figure 7 micromachines-12-00659-f007:**
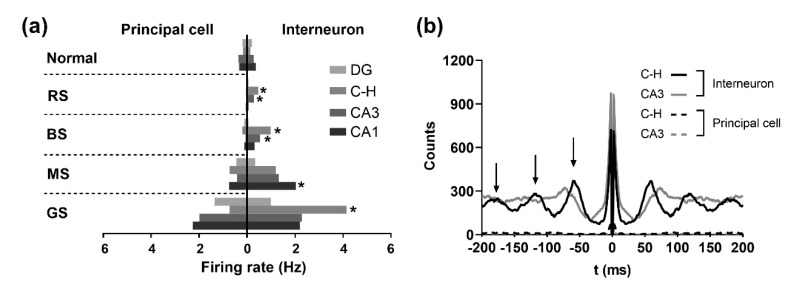
Comparison of spike firing rates and autocorrelograms of principal cells and interneurons. (**a**) Spike firing rates of principal cells and interneurons were dramatically increased after the induction of seizure. The firing rates were compared between principal cells and interneurons in the normal stage, resting state (RS), beginning state (BS), middle stage (MS), and grand mal stage (GS). (Principal cells: n = 16, 23, 17, and 12 in the DG, the hilus, CA3, and CA1; interneurons, n = 12, 9, 20, and 12 in the DG, the hilus, CA3, and CA1.) (**b**) Autocorrelograms of principal cells showed no oscillations, whereas those of interneurons showed rhythmic oscillations in the hilus and CA3 areas. Black arrows indicate the oscillation peaks. * *P* < 0.05 (interneurons vs. principal cells in same areas), ANOVA test followed by Tukey’s post hoc test. C-H: hilus.

**Figure 8 micromachines-12-00659-f008:**
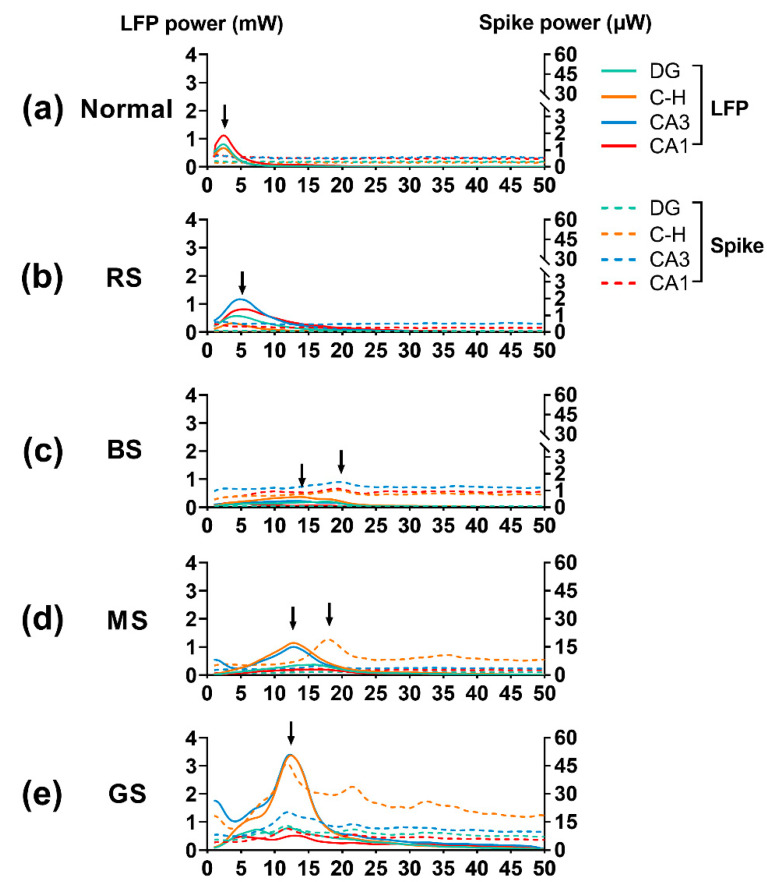
The joint analysis of power spectral density (PSD) of neural spikes and local field potentials (LFPs). (**a**) Normal stage. (**b**) Resting stage. (**c**) Beginning stage of seizure (BS). (**d**) Middle stage of seizure (MS). (**e**) Grand mal stage of seizure (GS). C-H: hilus.

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
