# Peer review of "In Vivo Microelectrode Arrays for Detecting Multi-Region Epileptic Activities in the Hippocampus in the Latent Period of Rat Model of Temporal Lobe Epilepsy"

_micromachines, 2021, doi:10.3390/mi12060659_

Round 1

Reviewer 1 Report

The author reported fabrication of microelectrode arrays to detect the neural activities in multiple hippocampal regions in the latent period of Temporal lobe epilepsy. The manuscript lacks few essential experiments. The in vivo performed in a complex matrix hence it requires a controlled and an optimized experimental conditions prior implantation therefore, at this stage the manuscript is not recommended for publication due to insufficient Data. The details of these experiments are appended below

  • The author electrodeposited platinum nanoparticles (PtNPs) to decrease the electrode impedance however, detail of these PtNPs are not mentioned in the manuscript.
  • Fabrication of MEA was done using different kind of materials however, it is not clear what are the benefit of each material and why been chosen to fabricate MEA?
  • Selectivity and the interference study were not reported.

Author Response

Reply to Reviewer 1:

Comments of Reviewer 1:

The author reported fabrication of microelectrode arrays to detect the neural activities in multiple hippocampal regions in the latent period of Temporal lobe epilepsy. The manuscript lacks few essential experiments. The in vivo performed in a complex matrix hence it requires a controlled and an optimized experimental conditions prior implantation therefore, at this stage the manuscript is not recommended for publication due to insufficient Data. The details of these experiments are appended below.

Point 1

The in vivo performed in a complex matrix hence it requires a controlled and an optimized experimental conditions prior implantation therefore, at this stage the manuscript is not recommended for publication due to insufficient Data.

Response 1

Thank you for raising the critical issue and it is an important comment. We added the detailed description of the animal breeding conditions, experimental conditions for in vivo detection, and the parameter settings for signal recording.

  • All rats were housed individually under controlled environment with a 12-hour light/dark cycle. Temperature and humidity were maintained at 22 ± 2°C and 40 ± 5%, respectively. (Page 4, line 127-129)
  • Subject rats were fixed on the stereotaxic device and received craniotomies under isoflurane anesthesia throughout the procedure. The concentration of isoflurane was 4­–5% for induction and 0.5–2%­­­ for maintenance of general anesthesia. At the same time, the heating pad of 37°C was used to maintain body temperature of rats. After the craniotomy, the MEA was slowly implanted into hippocampus (AP -3.6 mm) from left hemisphere according to the Rats Atlas of Paxinos and Watson. (Page 4, line 139-144)
  • Before the signal recording, the instrument should run for at least 10 min until the background noise was stable. The threshold for neural spike detection was set to 3 times maximum amplitude of background noise. Neural spikes were then extracted by high-pass filter (> 250 Hz) and local field potentials (LFPs) were obtained by low-pass filter (0–250 Hz). (Page 5, line 157-161)

More importantly, surface impedance of detecting sites was the essential and decisive parameter for electrophysiological detection. Because the neural signals are generally at the micro amplitude level, the low impedance was needed to improve signal-to-noise ratio. In the present study, the electrochemical impedance spectroscopy (EIS) test was used to assess the impedance of MEAs. The results showed that the average impedance magnitude at 1 kHz of detecting sites was 62.30 ± 3.28 Ω/μm2 and the phase angle was –13.93 ± 1.36°, which have been proven to be optimized for detecting neural signals. In addition, the recording instrument and the skull of rats were connected with ground wire to shield environmental noise. And the recording system could shield powerline interference through band stop filter.

In conclusion, the experimental condition and performance parameter of MEAs were optimized for detecting neural signals in the present study.

References

  1. Fan, X.; Song, Y.; Ma, Y.; Zhang, S.; Xiao, G.; Yang, L.; Xu, H.; Zhang, D.; Cai, X. In Situ Real-Time Monitoring of Glutamate and Electrophysiology from Cortex to Hippocampus in Mice Based on a Microelectrode Array. Sensors 2017, 17, 8, doi:10.3390/s17010061.
  2. Song, Y.; Xiao, G.; Li, Z.; Gao, F.; Wang, M.; Xu, S.; Cai, X. Electrophysiological Detection of Cortical Neurons under Gamma-Aminobutyric Acid and Glutamate Modulation Based on Implantable Microelectrode Array Combined with Microinjection. Conf. Proc. IEEE Eng. Med. Biol. Soc. 2018, 2018, 4583-4586, doi:10.1109/EMBC.2018.8513118

Point 2

The author electrodeposited platinum nanoparticles (PtNPs) to decrease the electrode impedance however, detail of these PtNPs are not mentioned in the manuscript.

Response 2

Thank you for raising this issue and it is an important and valuable comment. We added the detailed description of PtNPs modification and the explanation of using Pt to form conducting layer in the revised manuscript. Relevant references were also cited in the manuscript.

  • In present study, the platinum was used for conducting interface of MEAs due to its chemical stability, good biocompatibility, and high conductivity. (Page 3, line 101-102.)
  • To decrease the impedance of electrodes and enhance the signal-noise ratio for recording, detecting sites were electroplated with platinum nanoparticles (PtNPs) via chronoamperometry. PtNPs could enlarge the specific surface area and enhance the electron transmission capabilities of detecting sites. In addition, the electroplated PtNPs is characterized by chemical stability and strong mechanical, which is conducive to stable and durable electrophysiological detection. (Page 4, line 113-118)
  • The electroplating was carried out at −1.2 V for 50 s using the three-electrode system with Ag/AgCl reference electrode and Pt counter electrode. (Page 4, line 120-121)

References

  1. Fan, X.; Song, Y.; Ma, Y.; Zhang, S.; Xiao, G.; Yang, L.; Xu, H.; Zhang, D.; Cai, X. In Situ Real-Time Monitoring of Glutamate and Electrophysiology from Cortex to Hippocampus in Mice Based on a Microelectrode Array. Sensors 2017, 17, 8, doi:10.3390/s17010061.
  2. Ferro, M.D.; Melosh, N.A. Electronic and Ionic Materials for Neurointerfaces. Adv. Funct. Mater. 2018, 28, doi:10.1002/adfm.201704335.
  3. Song, E.; Li, J.; Won, S.M.; Bai, W.; Rogers, J.A. Materials for flexible bioelectronic systems as chronic neural interfaces. Nat. Mater. 2020, 19, 590-603, doi:10.1038/s41563-020-0679-7.
  4. Guarnieri, V.; Biazi, L.; Marchiori, R.; Lago, A. Platinum metallization for MEMS application. Biomatter 2014, 4, e28822, doi:10.4161/biom.28822.

Point 3

Fabrication of MEA was done using different kind of materials however, it is not clear what are the benefit of each material and why been chosen to fabricate MEA?

Response 3

Thank you for raising this issue and it is an important and valuable comment. We added the explanation of using these materials in the preparation of MEAs in the revised manuscript. Relevant references were also cited in the manuscript.

  • Silicon is characterized by low cytotoxicity and excellent biocompatibility. Micro-scale silicon performed good toughness and could withstand the stress of brain tissue during the in vivo detection. (Page 3, line 91-92)
  • Initially, the surface of SOI wafer was thermally oxidized to produce thin SiO2 film (Figure 2a) to insulate the surface of SOI wafer. (Page 3, line 96-97)
  • The conducting layer was composed of Titanium/Platinum (Ti/Pt, 30 nm/250 nm) and was deposited upon the SiO2 followed by the lift-off processing to form detecting sites, bonding pads and micro wires (Figure 2c). In present study, the platinum was used for conducting interface of MEAs due to its chemical stability, good biocompatibility, and high conductivity. And the titanium layer was deposited to enhance the adhesion between the Pt layer and the SiO2 (Page 3, line 98-103)
  • The strategy of co-deposition of Si3N4 and SiO2 was carried out to prevent the bending of MEAs due to the mechanical stress. (Page 4, line 106-107)

References

  1. Fekete, Z. Recent advances in silicon-based neural microelectrodes and microsystems: a review. Sens. Actuators, B 2015, 215, 300-315, doi:https://doi.org/10.1016/j.snb.2015.03.055.
  2. Ferro, M.D.; Melosh, N.A. Electronic and Ionic Materials for Neurointerfaces. Adv. Funct. Mater. 2018, 28, doi:10.1002/adfm.201704335.
  3. Song, E.; Li, J.; Won, S.M.; Bai, W.; Rogers, J.A. Materials for flexible bioelectronic systems as chronic neural interfaces. Nat. Mater. 2020, 19, 590-603, doi:10.1038/s41563-020-0679-7.
  4. Zhang, S.; Song, Y.; Wang, M.; Zhang, Z.; Fan, X.; Song, X.; Zhuang, P.; Yue, F.; Chan, P.; Cai, X. A silicon based implantable microelectrode array for electrophysiological and dopamine recording from cortex to striatum in the non-human primate brain. Biosens. Bioelectron. 2016, 85, 53-61, doi:10.1016/j.bios.2016.04.087.

Point 4

Selectivity and the interference study were not reported.

Response 4

Thank you for raising this issue and it is a valuable comment.

In the present study, the electrophysiological signals including neural spikes and local field potentials were mainly recorded. After the MEA was implanted into the brain, the detecting sites were attached closely to the neurons. When neurons generate action potentials, the signals propagate down the neurons as current which flows in and out of the cell through excitable membrane. The flowing current results in the variations of electric field around neurons, which could be detected by microelectrodes in the form of discharge signals. Technically, neural spikes represent the action potentials discharged by neurons and were extracted from detected signals by high-pass filter, while local field potentials represent discharging dynamics of local neuron population and were extracted by low-pass filter. Meanwhile, the extracting threshold was set to 3 times maximum amplitude of background noise. Thus, the prepared MEAs performed with high selectivity for neural spikes and local field potentials in the present study.

In addition, because the MEAs were implanted deeply into the brain tissue, the signal detection would not be interfered by the electromyographic noise. And the interferences during in vivo detection were generally derived from powerline interference and environmental noise. In our research, the recording instrument and the skull of rats were connected with ground wire to shield environmental noise. And the recording system could shield powerline interference through band stop filter. Meanwhile, the threshold for neural signal detection was set to 3 times maximum amplitude of background noise. The above strategies can greatly improve the signal-to-noise ratio of MEAs and eliminate the interference of noise.

References

  1. Buzsaki, G.; Anastassiou, C.A.; Koch, C. The Origin of Extracellular Fields and Currents--EEG, ECoG, LFP and Spikes. Nat. Rev. Neurosci. 2012, 13, 407-420, doi:10.1038/nrn3241.
  2. Harvey, V.L.; Dickenson, A.H. Extracellular Recording. In Encyclopedia of Psychopharmacology, Stolerman, I.P., Ed.; Springer Berlin Heidelberg: Berlin, Heidelberg, 2010; pp. 522-525.

Reviewer 2 Report

This is a very interesting paper that involves the customization of a multielectrode array based on the biological question, namely the manifestation of seizure induction at the cellular (neuronal) level within distinct regions of the brain. While the customized array is, in a sense, the micromachine, the paper provides the requisite detail on the fabrication of the device. Moreover, it is rich with experimental observations which will bring new readers to the journal to appreciate a study that has both engineering and neuroscience novelty. 

There really are no major concerns with the manuscript. Instead, there are a few places where English needs improvement or there needs to be clarity to more readily convey the thought. A good editing should take care of most of the minor issues. Instead of belaboring those points, I will offer the following more substantive suggestions that go beyond grammar and style.

  1. page 1, lines 29-30: edit "capable to detect neural activities at single-cell level... brain diseases" to "capable of detecting neural activity at the single-cell level and are widely applied to the study of the functional circuitry of neuronal networks in health and disease".
  2. page 1, line 41: edit "can go through" to "is characterized by".
  3. page 2, line 47: edit "optima MEA" to "customized design of a MEA optimized"
  4. page 2, line 48: edit "difference of the neural activities" to "neural dynamics".
  5. page 2, lines 57-58: delete "This would ... regulation of TLE". This is somewhat overstated.
  6. page 4, lines 144-145: edit "histochemistry... trace of MEA" to "histochemical analyses of the tissue implanted with the MEA".
  7. page 6, line 189-190: "gradually drastic fluctuations" -- edit for clarity
  8. The authors refer to electrode impedance and phase, which is technically incorrect. What they mean is electrode impedance magnitude and phase, as impedance is composed of both magnitude and phase components. Please edit text and figure legends throughout.
  9. page 9, lines 262-265: "Due to the synchronized frequency band ... hilus and CA3 areas" - Please edit for clarity.
  10. page 11, lines 324-326: "In present study ... regulation of epilepsy" - this sentence was very unclear and needs to be edited for clarity. Is there such a thing as "accurate regulation of epilepsy"

Author Response

Reply to Reviewer 2:

Comments of Reviewer 2:

This is a very interesting paper that involves the customization of a multielectrode array based on the biological question, namely the manifestation of seizure induction at the cellular (neuronal) level within distinct regions of the brain. While the customized array is, in a sense, the micromachine, the paper provides the requisite detail on the fabrication of the device. Moreover, it is rich with experimental observations which will bring new readers to the journal to appreciate a study that has both engineering and neuroscience novelty. 

There really are no major concerns with the manuscript. Instead, there are a few places where English needs improvement or there needs to be clarity to more readily convey the thought. A good editing should take care of most of the minor issues. Instead of belaboring those points, I will offer the following more substantive suggestions that go beyond grammar and style.

Point 1

page 1, lines 29-30: edit "capable to detect neural activities at single-cell level... brain diseases" to "capable of detecting neural activity at the single-cell level and are widely applied to the study of the functional circuitry of neuronal networks in health and disease".

Response 1

Thanks for your valuable suggestion and we revised the manuscript according to your advice.

  • Implantable microelectrode arrays (MEAs) are capable of detecting neural activity at single-cell level and are widely applied to the study of the functional circuitry of neuronal networks in health and disease. (Page 1, line 29-31)

Point 2

page 1, line 41: edit "can go through" to "is characterized by".

Response 2

Thanks for your valuable suggestion and we revised the sentence in the manuscript.

  • The evolution of TLE is characterized by several stages, including the latent period and the chronic period. (Page 1, line 41-42)

Point 3

page 2, line 47: edit "optima MEA" to "customized design of a MEA optimized".

Response 3

Thanks for your valuable suggestion and we revised the sentence in the manuscript.

  • the customized design of a MEA optimized for the multi-region in vivo detection is urgently needed to explore the neural dynamics between the hippocampal subregions in this period. (Page 2, line 48)

Point 4

page 2, line 48: edit "difference of the neural activities" to "neural dynamics".

Response 4

Thanks for your valuable suggestion and we revised the manuscript according to your advice.

  • the customized design of a MEA optimized for the multi-region in vivo detection is urgently needed to explore the neural dynamics between the hippocampal subregions in this period. (Page 2, line 49)

Point 5

page 2, lines 57-58: delete "This would ... regulation of TLE". This is somewhat overstated.

Response 5

Thanks for your valuable suggestion and we deleted this sentence in the revised manuscript according to your advice.

Point 6

page 4, lines 144-145: edit "histochemistry... trace of MEA" to "histochemical analyses of the tissue implanted with the MEA".

Response 6

Thanks for your valuable suggestion and we revised the manuscript according to your advice.

  • Recorded neural signals were analyzed after in vivo experiments, and rats were received transcardial perfusion of paraformaldehyde (PFA, 4%) followed by histochemical analyses of the tissue implanted with the MEA. (Page 5, line 163-164)

Point 7

page 6, line 189-190: "gradually drastic fluctuations" -- edit for clarity.

Response 7

Thank you for raising this issue and it is an important and valuable suggestion. We revised the description in the manuscript as following:

  • Spinous waves in LFPs disappeared in the BS and the amplitude of LFPs were significantly increased in the MS. (Page 6, line 213-214)

Point 8

The authors refer to electrode impedance and phase, which is technically incorrect. What they mean is electrode impedance magnitude and phase, as impedance is composed of both magnitude and phase components. Please edit text and figure legends throughout.

Response 8

Thank you for raising this issue and it is an important and valuable suggestion. We revised the text and figures as well as figure legends about impedance test of MEAs in the manuscript.

  • The result showed that modification with PtNPs reduced the impedance magnitude (Figure 4a) and improved phase shift of detecting sites (Figure 4b). The average impedance magnitude at 1 kHz of detecting sites was decreased from 7125.89 ± 900.7 Ω/μm2 to 62.30 ± 3.28 Ω/μm2 (Figure 4c), and the phase angle was decreased from –72.98 ± 1.65° to –13.93 ± 1.36° (Figure 4d). (Page 6, line 199-203)

Figure 4. Impedance magnitude and phase assessment. (a) and (b) Electrode impedance magnitude and phase distribution before and after PtNPs modification. (c) The average impedance magnitude of electrode at 1kHz decreased from 7125.89 ± 900.7 Ω/μm2 to 62.30 ± 3.28 Ω/μm2. (d) The phase changed from -72.98 ± 1.65° to -13.93 ± 1.36°. ***P < 0.001, paired t-test, n = 10 for each group. (Page 6, line 206-209)

Point 9

page 9, lines 262-265: "Due to the synchronized frequency band ... hilus and CA3 areas" - Please edit for clarity.

Response 9

Thank you for raising this issue and it is a valuable suggestion. We revised the sentence in the manuscript as following:

  • Due to the synchronized frequency band (9–15 Hz) of LFP and spike was approximate to the oscillation frequency (about 15 Hz) of interneurons, it implicated that dynamics of neuron population were mainly dominated by rhythmic oscillations of interneurons in the hilus and CA3 areas, which might cause the formation and spread of seizure activities. (Page 9, line 283-287)

Point 10

page 11, lines 324-326: "In present study ... regulation of epilepsy" - this sentence was very unclear and needs to be edited for clarity. Is there such a thing as "accurate regulation of epilepsy".

Response 10

Thank you for raising this issue and it is a valuable suggestion. We revised the sentence in the manuscript and the relevant references were also cited.

  • In present study, the dysfunction in hippocampal subregions was successfully detected at cellular level, which conduced to the prediction of the epileptic formation and the accurate diagnosis of the seizure focus. Furthermore, studies have been devoted to the targeted and closed-loop regulation of epilepsy in recent years [24,25], which were highly dependent on the accurately detecting of epileptic focus. The fabricated MEAs and novel findings in present study could provide an important basis to develop targeted therapy of epilepsy in the future. (Page 11, line 348-354)

References

  1. Sisterson, N.D.; Wozny, T.A.; Kokkinos, V.; Constantino, A.; Richardson, R.M. Closed-Loop Brain Stimulation for Drug-Resistant Epilepsy: Towards an Evidence-Based Approach to Personalized Medicine. Neurotherapeutics 2019, 16, 119-127, doi:10.1007/s13311-018-00682-4.
  2. Berényi, A.; Belluscio, M.; Mao, D.; Buzsáki, G. Closed-Loop Control of Epilepsy by Transcranial Electrical Stimulation. Science 2012, 337, 735, doi:10.1126/science.1223154.

Round 2

Reviewer 1 Report

The author has answered the concern comments adequately and supplied suffiecient explamnation. Now the manuscript can be considered for publication.